# An Efficient Volume Integral Equation Method for Analysis of Boresight Error of a Radome with Minor Ablation

**Xiao-Yang He** [ID]**, De-Hua Kong, Wen-Wei Zhang and Ming-Yao Xia *** [ID]

School of Electronics, Peking University, Beijing 100871, China
* Correspondence: myxia@pku.edu.cn

**Abstract:** In this paper, an efficient method based on volume integral equation is developed to analyze the effects of ablation of a radome on the boresight error. To avoid recalculating the whole impedance matrix when the permittivity of the radome or the shape of the top portion is slightly changed due to ablation, the radome is divided into unaffected and affected parts and the volume equivalent current instead of the displacement current is used as the unknown. This permits us to reassemble rather than recalculate the impedance matrix when the ablation condition is altered. Moreover, a viable preconditioning technique is introduced and integrated with the multilevel fast multipole algorithm (MLFMA) to cope with the electrically large antenna-radome system (ARS). Simulation results are provided for the boresight error (BSE) and boresight error slope (BSES) of the ARS at some different ablation states. The present approach is considerably faster than using the conventional methods.

**Keywords:** ablation; antenna-radome system (ARS); boresight error (BSE); multilevel fast multipole algorithm (MLFMA); volume integral equation (VIE)

## 1. Introduction

The antenna-radome system (ARS) plays an indispensable role in many fields [1]. Traditionally, ARS is usually designed for stable environments. However, due to the influence of environment changes, such as temperature and external forces, the permittivity of the radome may vary in a narrow range and a little defection of the top portion may happen. For example, ablation of the radome is common for supersonic vehicles flying in the high atmosphere [2]. The performance of the antenna in ARS may be influenced a lot by ablation, including the radiation pattern (RP), boresight error (BSE) and boresight error slope (BSES). Therefore, accurately analyzing the effects of ablation on the performance is of great importance for practical applications.

The concept of BSE is described in [3], which describes the shift from the real target location when processed by the antenna looking through the radome. In recent years, there exists an increasing concern about the BSE problem of ARS. Many methods including geometrical optics (GO) [4], ray tracing (RT) [5,6], method of moments (MoM), multi-level fast multipole algorithm (MLMFA) [7,8] and some hybrid methods [9,10] have been employed to analyze the ARS problems. However, GO- and RT-related methods are not accurate enough especially for inhomogeneous radomes, while the common MLFMA accelerated MoM are not efficient for problems with changing permittivity and shape. In consideration that an MLFMA-accelerated volume integral equation (VIE) method should be one of the most suitable candidates [11–15], we are motivated to develop a tailored scheme under the MLFMA-VIE framework if either the permittivity, shape or both, are time-varying.

In this paper, an efficient method is proposed by taking full advantage of the results obtained at the initial moment, in order that many calculations can be bypassed at subsequent moments, including the filling of a large part of the impedance matrix elements

and acquiring of the preconditioner. The program is written in the Julia language, paralleled with multi-threading techniques, and is successfully applied to simulate a real-sized ARS discretized with millions of unknowns. The RPs of the ARS under different ablation states are calculated, and then the BSE and BSES are examined to investigate the degree of influence due to ablation.

## 2. Volume Integral Equation (VIE) Method

For MoM, the VIE of EFIE is expressed as

$$\boldsymbol{E}_i(\boldsymbol{r}) + \boldsymbol{E}_s(\boldsymbol{r}) = \boldsymbol{E}(\boldsymbol{r}), \ \boldsymbol{r} \in V \tag{1}$$

where $\boldsymbol{E}_i(\boldsymbol{r})$, $\boldsymbol{E}_s(\boldsymbol{r})$ and $\boldsymbol{E}(\boldsymbol{r})$ are the incident, scattering and total electric field at $\boldsymbol{r}$, respectively, and $V$ denotes the volume region. Moreover, $\boldsymbol{E}_s(\boldsymbol{r})$ can be expressed by a volume equivalent current $\boldsymbol{J}_V(\boldsymbol{r}')$ according to the equivalence principle.

$$\boldsymbol{E}_s(\boldsymbol{r}) = \eta_0 \boldsymbol{L}\big[\boldsymbol{J}_V(\boldsymbol{r}')\big] \tag{2}$$

where $\eta_0 = \sqrt{\mu_0/\varepsilon_0}$ is the characteristic impedance of background space, and the operator $\boldsymbol{L}$ is defined as

$$\boldsymbol{L}\big[\boldsymbol{X}(\boldsymbol{r}')\big] = -jk\left(1 + \frac{1}{k^2}\nabla\nabla\cdot\right)\int_V G(R)\boldsymbol{X}(\boldsymbol{r}')dV' \tag{3}$$

in which $k$ is the wave number of background space, and $G(R)$ is the Green function $G(R) = e^{-jkR}/(4\pi R)$ with $R = |\boldsymbol{r} - \boldsymbol{r}'|$.

After choosing suitable basis functions and weighting functions to discretize Equation (1), a matrix equation is obtained

$$\mathbb{Z}\mathbb{I} = \mathbb{V} \tag{4}$$

where $\mathbb{Z}$ is called the impedance matrix, $\mathbb{I}$ is a column vector containing the expansion coefficients of the unknown current and $\mathbb{V}$ is the excitation vector. Specifically,

$$Z_{mn} = \int_{V_n} \boldsymbol{w}_m(\boldsymbol{r}) \cdot \big\{\boldsymbol{E}(\boldsymbol{r}) - \eta_0 \boldsymbol{L}\big[\boldsymbol{J}_V(\boldsymbol{r}')\big]\big\}dV \tag{5}$$

$$V_m = \int_{V_m} \boldsymbol{w}_m(\boldsymbol{r}) \cdot \boldsymbol{E}^i(\boldsymbol{r})dV \tag{6}$$

where $\boldsymbol{w}_m(\boldsymbol{r})$ is the weighting function, and $\boldsymbol{J}_V(\boldsymbol{r}')$ and $\boldsymbol{E}(\boldsymbol{r})$ are determined through expanding them using basis functions.

There are generally two ways to solve Equation (4), namely, direct solvers and iterative solvers. For electrically large problems, iterative solvers are usually used because their computational complexity is relatively lower. Among them, the generalized minimum residual method (GMRES) is usually preferred for its optimal performance in most problems [16]. Besides, direct solvers require storing a full impedance matrix, which is prohibitive for an electrically large problem. Furthermore, for better convergence performance, a preconditioner such as sparse approximate inverse (SAI) is usually adopted in iterative solvers [17,18]. Once the matrix equation is solved and the equivalent current is found, post-processing such as RP and radar cross section (RCS) are examined.

Traditionally, to simulate ARS with ablation on a radome, the process described above needs to be solved at every moment, including the recalculations of the impedance matrix and preconditioner time by time, which is very inefficient. Therefore, an alternative approach is introduced below, which takes full advantage of the impedance matrix and preconditioner at the initial moment.

### 3. Modification for Changing Permittivity and Shape

The consequences of ablation on the radome include defecting on the radome and altering of permittivity. The two situations may be handled separately. In this paper, a real-sized radome given by Wang in [4] is used, as shown in Figure 1a.

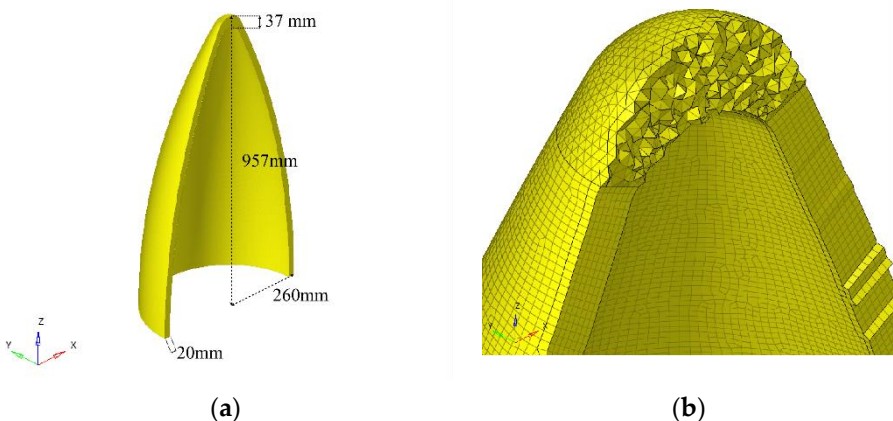

**Figure 1.** A real-sized radome model from [4] is used in this paper: (**a**) The shape of the radome with sizes; (**b**) the meshes of radome with mixed tetrahedra and hexahedra.

To accurately fit the ablation interface and reduce the unknowns, the top portion of the radome that may be ablated is meshed by tetrahedrons, while the remaining parts are meshed by hexahedrons, as shown in Figure 1b. Moreover, piecewise constant (PWC) basis functions [19] are used to expand the currents, and the Galerkin test is used to generate the impedance matrix. The PWC basis function is defined as

$$f_n(\boldsymbol{r}) = \left\{ \begin{array}{l} \hat{\boldsymbol{e}}_{x|y|z}, \boldsymbol{r} \in V_t \\ 0, else \end{array} \right. \tag{7}$$

where $\hat{\boldsymbol{e}}$ denotes the unit vector. Each tetrahedron or hexahedron contains 3 PWC basis function in $x$, $y$ and $z$ directions, respectively. For better convergence and close relationship with the impedance matrix [18], SAI is chosen to construct the preconditioner.

### 3.1. Consideration for Changing Permittivity

Traditionally, for inhomogeneous problems, the displacement current $\boldsymbol{J}_D = j\omega \boldsymbol{D}$ is taken as the unknown function to be expanded by using the linear SWG basis functions, which automatically meet the continuity requirement for $\boldsymbol{D}$ on common faces of meshes [20,21]. In this way, the electric field $\boldsymbol{E}$ and volume equivalent current $\boldsymbol{J}_V$ are expressed as $\boldsymbol{E} = \boldsymbol{J}_D / [j\omega\varepsilon(\boldsymbol{r})]$ and $\boldsymbol{J}_V = j\omega[(\varepsilon(\boldsymbol{r}) - \varepsilon_0)\boldsymbol{E} = [1 - \varepsilon_0/\varepsilon(\boldsymbol{r})]\boldsymbol{J}_D$. Unfortunately, when they are inserted into Equation (2), all the matrix elements need to be recalculated again and again when the $\varepsilon(\boldsymbol{r})$ of the radome is time-changing.

To avoid the situation mentioned above, we expand $\boldsymbol{J}_V$ and $\boldsymbol{E}$ as

$$\boldsymbol{J}_V(\boldsymbol{r}) = \sum_{n=1}^{N} I_n \boldsymbol{f}_n(\boldsymbol{r}), \ \boldsymbol{E}(\boldsymbol{r}) = \frac{1}{j\omega} \frac{\boldsymbol{J}_V(\boldsymbol{r})}{\varepsilon(\boldsymbol{r}) - \varepsilon_0} \tag{8}$$

Substituting these into Equation (5), we obtain the matrix element as $Z_{mn} = Z_{mn}^e(\varepsilon_n) + Z_{mn}^s$, which corresponds to the two terms in Equation (5). It is obvious that $Z_{mn}^s$ has nothing to do with $\varepsilon(\boldsymbol{r})$ regardless of whether it changes or not, while $Z_{mn}^e(\varepsilon_n)$ is changed only when involved in the region that $\varepsilon(\boldsymbol{r})$ is altered. As a result, we can write the matrix $\mathbb{Z}$ as

$$\mathbb{Z} = \mathbb{Z}^e + \mathbb{Z}^s \tag{9}$$

where $\mathbb{Z}^s$ is a dense matrix but needs to be calculated only once, while $\mathbb{Z}^e$ is a very sparse band matrix and only a small part of it needs to be recalculated when the permittivity is changed.

### 3.2. Consideration for Defect Due to Ablation

All basis functions are marked as un-defected in the initial state. Subsequently, the basis functions related to defection at a specified moment are marked as defected. The matrix $\mathbb{Z}$ can be written as

$$\mathbb{Z}_{N \times N} = \begin{bmatrix} \mathbb{Z}_{N^r \times N^r}^{rr} & \mathbb{Z}_{N^r \times N^d}^{rd} \\ \mathbb{Z}_{N^d \times N^r}^{dr} & \mathbb{Z}_{N^d \times N^d}^{dd} \end{bmatrix} \tag{10}$$

where the superscripts $r$ and $d$ stand for the remaining and defected regions, respectively; and $N^r$ and $N^d$ are the number of basis functions of the two regions, respectively. Usually, $N^r \gg N^d$ in practice. The three sub-matrixes $\mathbb{Z}_{N^r \times N^d}^{rd}$, $\mathbb{Z}_{N^d \times N^r}^{dr}$ and $\mathbb{Z}_{N^d \times N^d}^{dd}$ that are involved in the defected region should be discarded. However, doing so will destroy the structure of $\mathbb{Z}$ and we cannot reuse it at another moment or ablation condition. We may allocate another matrix to store $\mathbb{Z}^{rr}$, but the increased memory requirement may be unavailable.

To keep $\mathbb{Z}$ intact and without requiring extra memory, we write $\mathbb{I} = \begin{bmatrix} \mathbb{I}^r \mathbb{I}^d \end{bmatrix}^T$ and then assign $\mathbb{I}^d = \mathbf{0}$ before every iteration. The iteration process that involves the matrix-vector multiplication may be illustrated by

$$\mathbb{Z} \begin{bmatrix} \mathbb{I}^r \\ \mathbb{I}^d \end{bmatrix}_{\mathbb{I}^d = \mathbf{0}} = \begin{bmatrix} \mathbb{Z}^{rr} \mathbb{I}^r \\ \mathbb{Z}^{dr} \mathbb{I}^r \end{bmatrix} = \begin{bmatrix} \mathbb{F}^r \\ \mathbb{F}^d \end{bmatrix}_{\mathbb{F}^d = \mathbf{0}} \tag{11}$$

that is, we just need the part $\mathbb{F}^r$ and assign $\mathbb{F}^d = \mathbf{0}$ to update $\mathbb{I} = \begin{bmatrix} \mathbb{I}^r \mathbb{I}^d \end{bmatrix}^T$ and then assign $\mathbb{I}^d = \mathbf{0}$, and then assign $\mathbb{I}^d$ to start the next iteration.

### 3.3. Construction of Preconditioner

In practice, the time cost for acquisition of a preconditional matrix $\mathbb{P}$ usually takes more time than that to assemble the impedance matrix. Therefore, a fast-updating strategy for the preconditioner is required. For better manipulating of the preconditioner, the SAI method is preferred because its sparse pattern is the same as the impedance matrix of the near-interacting region $\mathbb{Z}_{near}$ in MLFMA.

For a defected radome due to ablation, the elements in preconditioner $\mathbb{P}$ are influenced by the basis functions in the defected region. However, they are far-neighbors in the leaf level of the octree, which could easily be efficiently updated by looping and re-calculating on far-neighbors of the defected cubes.

For changing of permittivity due to high temperature, the handling is more complicated. In SAI, $\mathbb{P}$ is calculated by minimizing

$$\left\| \mathbb{P} \mathbb{Z}_{near} - \mathbb{I} \right\|_{\min} \tag{12}$$

which means that a little perturbation on $\mathbf{Z}_{near}$ may lead to recalculation of all elements in $\mathbb{P}$. However, for our problem at hand, an efficient updating strategy can be devised. Suppose that the $\mathbb{P}$ at the initial stage has been obtained, which is approximately the inverse of $\mathbb{Z}_{near} = \mathbb{Z}^e + \mathbb{Z}_{near}^s$, i.e.,

$$\mathbb{P} \approx \left( \mathbb{Z}^e + \mathbb{Z}_{near}^s \right)^{-1} \tag{13}$$

where $\mathbb{Z}^e$ is the same as (5), which is a sparse band matrix, and $\mathbb{Z}_{near}^s$ is the near-interacting part of $\mathbb{Z}^s$, then, when the permittivity in some region is changed at a moment thereafter, the new preconditioner $\mathbb{P}'$ can be found by

$$\begin{aligned} \mathbb{P}' &\approx \left( \mathbb{Z}'^e + \mathbb{Z}_{near}^s \right)^{-1} = \left( \mathbb{P}^{-1} + \Delta \mathbb{Z}^e \right)^{-1} \\ &= \left[ \mathbb{P}^{-1} (\mathbb{I} + \mathbb{P} \Delta \mathbb{Z}^e) \right]^{-1} = (\mathbb{I} + \mathbb{I})^{-1} \mathbb{P} \\ &\approx \left[ (\mathbb{I} - \mathbb{I}) \sum_{m=0}^{\infty} \mathbb{I}^{2n} \right] \mathbb{P} \end{aligned} \tag{14}$$

where $\Delta\mathbb{Z}^e = \mathbb{Z}'^e - \mathbb{Z}^e$ and $\mathbb{X} = \mathbb{P}\Delta\mathbb{Z}^e$. Because $\mathbb{X}^{2n}$ decreases rapidly as $n$ increases, a few terms in the summation are sufficient. Both $\mathbb{P}$ and $\Delta\mathbb{Z}^e$ are sparse, and calculations of $\mathbb{X} = \mathbb{P}\Delta\mathbb{Z}^e$ and $\mathbb{X}^{2n}$ are very cheap. The new $\mathbb{P}'$ can be stored as the same pattern as $\mathbb{P}$ to share the memory.

### 3.4. Integration with MLFMA

MLFMA is an essential algorithm for analyzing electrically large problems. When the permittivity is changed, only the matrix $\mathbb{Z}^e$ will be changed, which are the near-interactions that do not relate to the far-region interaction in MLFMA. Thus, the original MLFMA program can be directly utilized without any modification for this case. If the shape is defected due to ablation, we can treat the case in the same way as described in Section 3.2, i.e., keeping the original matrix $\mathbb{Z}$ intact, but assigning those currents that are marked in the defected region to be zero in each iteration using MLFMA to accelerate the multiplication of a far-interacting matrix with a vector. The modification of codes to suit this case is very slight.

## 4. Numerical Results

In this section, a real-sized ablating ARS as shown in Figure 1 is simulated to validate the correctness of the proposed procedure. In all examples, the antenna array in the ARS is composed of $14 \times 14$ sized dipoles and works at 5 GHz. The excitation voltages of the array elements adopt the Taylor distribution, which generates patterns with low sidelobe [22]. The initial permittivity of the radome is taken to be $2.82(1 - 0.002j)$ and uniformly distributed.

All the programs in this paper are coded in the Julia language, which is an excellent language developed in recent years. It has comparable performance to C, C++ and Fortran, while its grammar is as concise as Python and Matlab [23]. These features free researchers from heavy coding work and make it easier to write high-performance programs. It is believed that Julia is going to be a rising star in scientific computing.

The platform used in this paper is an Ubuntu 2004 server with 40 cores Intel(R) Xeon(R) Gold 5215L CPU @ 2.50 GHz processor. The number of threads is set to 16 for better efficiency. The GMRES solver is provided by an open-source package released on GitHub [24].

### 4.1. Validation of the Present MoM (P-MoM)

It is important to validate the correctness of the proposed method before any further application. Here we compare the simulation results of an ARS with ablation using the conventional MoM and the present MoM for perturbating shape or permittivity. The maximum defection depth is set to 10 mm, and the permittivity changes in the range from $2.82(1 - 0.002j)$ to $2.96(1 - 0.01j)$. More details are provided in Section 4.3.

The sum and difference patterns of the ARS are calculated by MoM and P-MoM, and the normalized far-field patterns are shown in Figure 2. For these two cases, the patterns calculated by the MoM and P-MoM are found to be in good agreement. Moreover, in solving the MoM and P-MoM, the SAI and the present SAI (P-SAI) were adopted, respectively. The overlapped results shown in Figure 2 confirm the correctness and accuracy of the proposed method and programs.

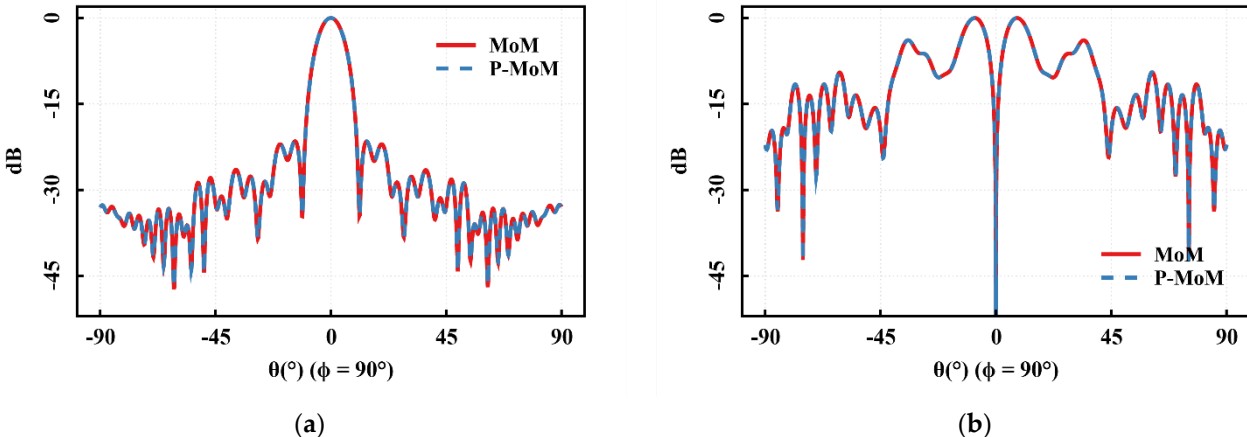

(**a**) (**b**)

**Figure 2.** Patterns of the ARS obtained by MoM and P-MoM are compared to demonstrate the correctness of P-MoM: (**a**) Normalized sum pattern; (**b**) normalized difference pattern.

### 4.2. Performance of the Present SAI (P-SAI)

The most important feature for a good preconditioner is to improve the convergence speed. For analyzing the convergence performance, the two examples in Figure 2 are used again. No preconditioner, SAI and P-SAI preconditioners are adopted in the solving process of P-MoM. Then, the relative residual errors are compared, which are defined as

$$\epsilon = \frac{\|\mathbb{Z}\mathbb{I} - \mathbb{V}\|_2}{\|\mathbb{V}\|_2} \tag{15}$$

As Figure 3 shows, in both graphs, P-SAI has the same accelerating performance as the traditional SAI. This proves that the approximation in P-SAI theory is applicable and effective in practice.

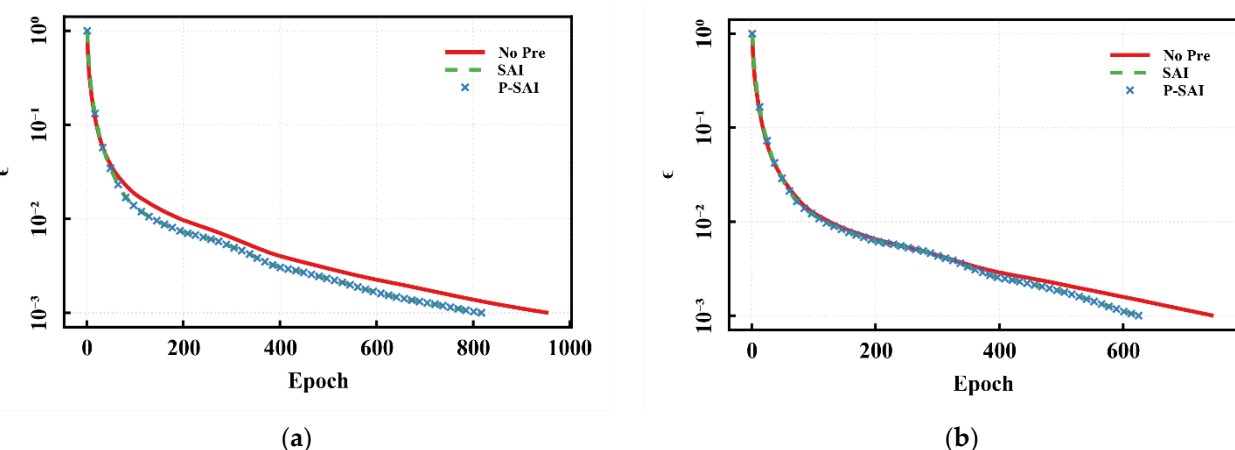

(**a**) (**b**)

**Figure 3.** The relative residual errors in convergence history of GMRES are recorded by P-MoM with no preconditioner, SAI preconditioner and P-SAI preconditioner. The same examples in Figure 2 are used here: (**a**) Convergence history in solving the sum pattern; (**b**) convergence history in solving the difference pattern.

Meanwhile, once the SAI is obtained at the initial moment, the time spent to construct the P-SAI in subsequent simulations is only about 35% of that in the traditional SAI. More detailed results about the time costs will be listed in the following section.

### 4.3. BSE of Ablating Radome

As the P-MoM has been proven to be a valid method to analyze electromagnetic performance of perturbation problems for changing permittivity/shape, we use it to simulate a

real-sized ablating ARS working at 5 GHz as mentioned above. It is meshed by 24,292 tetra-hedrons and 702,632 hexahedrons, which results in a total of 2,180,772 unknowns by using the PWC basis functions.

The simulation is carried out at five different moments, where t0 stands for the initial moment, and t1–t4 are ablating moments. The permittivity of the radome is uniform at the t0 moment, which is taken as $2.82(1 - 0.002j)$. At the t1, t2, t3 and t4 moments, the permittivity varies from the minimum $2.82(1 - 0.002j)$ at the bottom to the maximums $2.855(1 - 0.004j)$, $2.89(1 - 0.006j)$, $2.925(1 - 0.008j)$ and $2.96(1 - 0.01j)$, respectively, near the top.

Figure 4a shows the distribution of relative amplitude of permittivity on the radome, where 0 corresponds to the initial permittivity $2.82(1 - 0.002j)$ and 1 represents the maximum permittivity at the specified moment. For defecting on the top portion of the radome, the defection depths are set to be 1 mm, 4 mm, 7 mm and 10 mm from t1 to t4 as shown in Figure 4b. The value and distribution results of permittivity on the radome and defection depth are referred to in [4].

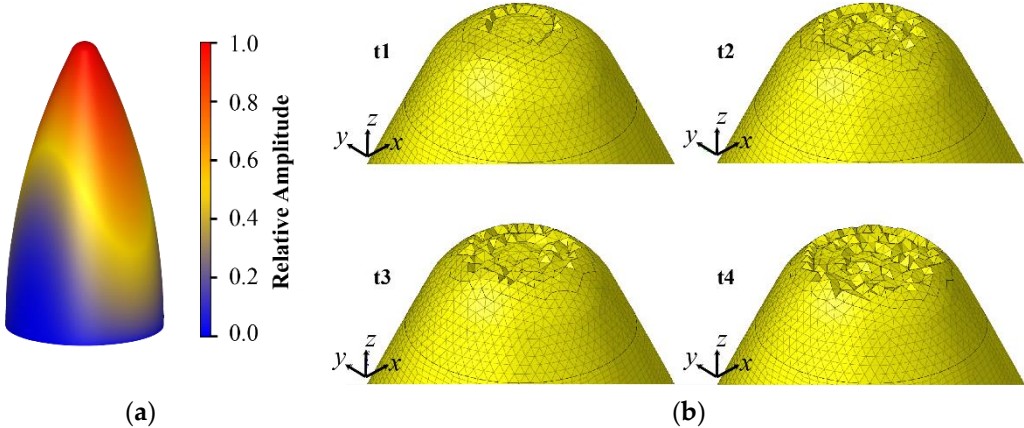

(**a**)　　　　　　　　　　　　　　　　　　　　　　　　　　(**b**)

**Figure 4.** Distribution of permittivity and defection on the radome for simulation of ablation process: (**a**) Relative amplitude of permittivity of the radome; (**b**) three-dimensional views of defection on the radome at four ablating moments.

At the five moments, the boresight of the antenna array rotates from $0°$ to $20°$ in plane $\phi = 90°$ to calculate the BSE, which is defined as the difference between the aiming angle of the antenna and ARS. BSES is defined as the slope of BSE. In a fixed meridian plane, say $\phi = 90°$, they are calculated by

$$
\begin{aligned}
BSE(\theta) &= f_r(\theta) - \theta \\
BSES(\theta) &= \frac{\partial}{\partial \theta} BSE(\theta) \approx \frac{\Delta BSE(\theta)}{\Delta \theta}
\end{aligned}
\tag{16}
$$

where $\theta$ is the intended aiming angle or scanning angle of the antenna array, and $f_r(\theta)$ is the real aiming angle of the ARS.

The numerical results of BSE and BSES are shown in Figure 5a,c, respectively. The BSE relative to no ablation (RBSE) is also shown in Figure 5b for comparison.

As shown in Figure 5a, without ablation (red line), the BSE increases as the aiming angle $\theta$ increases when $\theta < 5°$, then reaches a plateau between $5°$ and $10°$, then increases rapidly to $0.6°$ at $\theta = 17°$ and after that it begins to decline.

When ablation happens, the BSE follows the same tendency but with different degrees, which is clearly shown in Figure 5b, where the differences of each line minus the red line are plotted. The maximum difference is about $0.13°$ when $\theta = 18°$. This tiny difference of aiming angle may result in a missed distance of tens of meters for a target tens of kilometers away.

As shown in Figure 5c, the BSES varies sharply when $\theta < 5°$, which seems to be inaccurate because of insufficient resolution. After that the BSES varies smoothly.

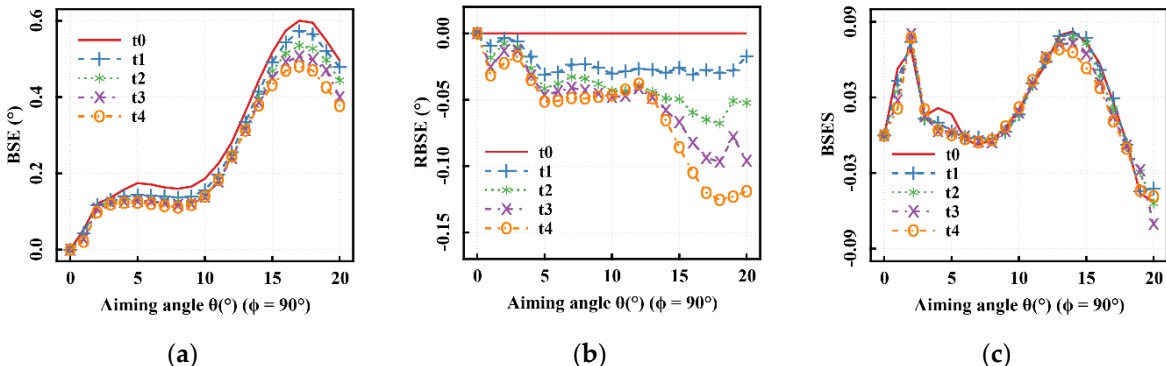

**Figure 5.** Five ablating moments of the ARS mentioned above are simulated using P-MoM. The BSE, RBSE and BSES at the five moments are calculated to examine the influence of ablation on ARS performance: (**a**) BSE; (**b**) RBSE; (**c**) BSES.

In Figure 6, the overall difference patterns for the antenna without radome (red line), antenna with intact radome (blue dot) and antenna with defected radome (green dot) at t4 and $\theta = 18°$ are displayed. A zoomed-in look of the valley region is given in the inset, which shows the obvious differences between one another. It can be seen from the inset that the ablation makes the zero-depth about 1 dB deeper and the BSE 0.13° smaller than the original ARS. These small variations are meaningful for real engineering applications.

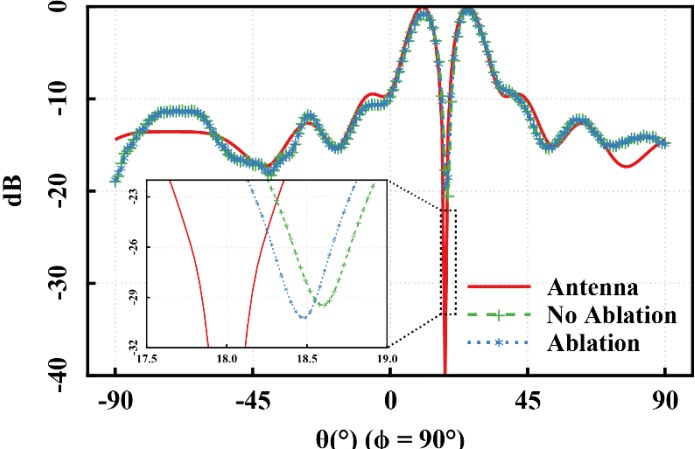

**Figure 6.** The overall difference patterns of ARS with and without ablation are compared when aiming angle $\theta = 18°$ and the local magnification graph near the valley of the pattern are also shown to make things clear.

### 4.4. Performance of P-MoM

The present MoM (P-MoM) and the traditional MoM require almost the same amount of core memory, thus we focus on comparing the CPU time. The time consumed by using MoM and P-MoM for the five ablating moments described in the last section are compared, as shown in Table 1.

At the first moment, P-MoM does the same things as the MoM, thus they take the same time. After that, the P-MoM will save about 99% and 65% time in constructing the impedance matrix and preconditioner, respectively, which are remarkable improvements. However, the overall speed up is only 27% because the CPU time to solve the matrix equation takes a large proportion of the total simulation time, which is nearly the same for the two approaches. More efforts are needed in follow-up work to reduce the memory requirement and increase the computational scale to meet the demand for higher operating frequency.

**Table 1.** The time consumption at different stages of the five moments using MoM and P-MoM.

| Moments | Stage | Time Consumption (s) | | Reduction |
|---|---|---|---|---|
| | | MoM | P-MoM | |
| t0 | Impedance Assembling | 196 | 196 | 0% |
| | Preconditioner Assembling | 2272 | 2272 | 0% |
| | Solving | 2523 | 2523 | 0% |
| t1 | Impedance Assembling | 196 | 1.6 | 99% |
| | Preconditioner Assembling | 2230 | 728 | 67% |
| | Solving | 2564 | 2575 | 0% |
| t2 | Impedance Assembling | 192 | 1.7 | 99% |
| | Preconditioner Assembling | 2214 | 801 | 64% |
| | Solving | 2503 | 2494 | 0% |
| t3 | Impedance Assembling | 193 | 1.7 | 99% |
| | Preconditioner Assembling | 2193 | 765 | 65% |
| | Solving | 2476 | 2491 | −1% |
| t4 | Impedance Assembling | 193 | 1.7 | 99% |
| | Preconditioner Assembling | 2166 | 761 | 65% |
| | Solving | 2451 | 2465 | −1% |
| **Total** | | **24,368** | **17,881.7** | **27%** |

## 5. Conclusions

In this paper, a viable procedure based on the MLFMA-VIE framework is presented to analyze the pointing error problem due to ablation of an antenna-radome system (ARS). The ablation may result in the change of permittivity of the radome because of high temperature and defect of the top portion because of burning-out. By using the volume equivalent current instead of the displacement current as the expanded unknown, recalculations of most impedance matrix elements are avoidable when the permittivity or shape is changing in a small range. As a result, a reduction of 99% for time for impedance matrix assembling and 65% for time for preconditioner constructing is achieved. A real-sized ablating ARS working at 5 GHz, having 2,180,772 unknowns, is analyzed to predict the boresight error (BSE) and boresight error slope (BSES), and the results seem reasonable, which should have important reference values for real engineering applications.

**Author Contributions:** Conceptualization, X.-Y.H. and M.-Y.X.; methodology, M.-Y.X.; validation, D.-H.K. and W.-W.Z.; formal analysis, X.-Y.H.; investigation, X.-Y.H., D.-H.K. and W.-W.Z.; data visualization; X.-Y.H. and D.-H.K.; writing—review and editing, X.-Y.H., M.-Y.X., D.-H.K. and W.-W.Z. All authors have read and agreed to the published version of the manuscript.

**Funding:** This research was supported by National Natural Science Foundation of China (NSFC) under grants 62171005 and 62231001.

**Conflicts of Interest:** The authors declare no conflict of interest.

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
