# Peer review of "An Efficient Volume Integral Equation Method for Analysis of Boresight Error of a Radome with Minor Ablation"

_electronics, doi:10.3390/electronics11233861_

Round 1

Reviewer 1 Report

1. What is the range of change in permittivity of the radome corresponding to temperature change? Does this paper cover all of this range?

2. Specific expressions of the basis and the weighting functions used in the calculation should be added in Section 2 or Section 3.

3. Please indicate that the two (MoM and P-MoM) overlapped in Fig. 2(a) and (b).

Reviewer 2 Report

In this paper, authors have described, in our opinion, an original methodology for analyzing the performance of antennas affected by the ablation of randome in supersonic vehicles flying in the high atmosphere.

The radiation pattern, boresight error, and other problems are the objectives of this work because they are veri impacted for this kind of mechanical problem.

As the influence of the ablation randome is located in a small part of the domain, the authors have developed a suitable methodology that corrects the changing of the physical properties in this region without to applied it in the entire domain.

Concerning mathematical formalism, the authors have created a creative way of detecting the elements affected by the ablation and combined it with an original preconditioner of the impedance matrix.  

Reviewer 3 Report

Engaging and well-written report on the ablation effect of radomes, for aerospatial applications. It focuses on the numerical solution. It is a complex and valid problem to tackle and is usually neglected by commercial solvers, so I guess it can be extended to other cases as well (ex. deformations on large reflector antennas). Some minor remarks:

-pag 162, Taylor instead of Tayler

- where does the radome material permittivity come from? Some known material?

- what is the input to the Julia Code? The 3D-CAD file of the radome? Has the structure already been meshed when imported?

- Fig. 2 both curves look the same to me. If so, there is no need to show them. If not, another parameter should be given to indicate maybe the worst-case difference across the band.

- authors use Julia, which is an emerging language. In a few years, it might become a big option or just disappears. So my point is authors could mention the reasons they've picked Julia instead of other alternatives (e.g. Python, C, Matlab etc).
